# Integrating Viral Infection and Correlation Analysis in *Passiflora edulis* and Surrounding Weeds to Enhance Sustainable Agriculture in Republic of Korea

**DOI:** 10.3390/v17030383

**Published:** 2025-03-07

**Authors:** Min Kyung Choi

**Affiliations:** Jeonbuk State Agricultural Research & Extension Service, Iksan 54591, Republic of Korea; mk8911@korea.kr; Tel.: +82-63-290-6061

**Keywords:** *Passiflora edulis*, viral infection, sustainable agriculture, PCR diagnostics, virus transmission control

## Abstract

*Passiflora edulis*, introduced to the Republic of Korea in 1989 and commercially cultivated since 2012, has faced recent challenges due to viral infections impacting growth, yield, and quality. This study aimed to investigate the viral infections in *P. edulis* and surrounding weeds at cultivation sites in the Republic of Korea, examining possible correlations between the infections for sustainable agriculture. Over five years, *P. edulis* and weed samples were collected for virus diagnosis using PCR and RT-PCR assays, analyzing the infection status in both *P. edulis* and weeds and across weed species/families. The findings revealed infections with EuLCV, PaLCuGdV, CMV, and EAPV in both *P. edulis* and weeds, with PaLCuGdV showing the highest infection rate. Although no direct correlation was found between the presence of the same viruses in *P. edulis* and weeds, suggesting that there may be interactions among different viruses, the study highlighted that EuLCV infection could exacerbate symptoms when coinfected by other viruses. The study underscores the importance of implementing preventive measures within greenhouses to control virus transmission, offering insights for strategic management of viral diseases in *P. edulis* cultivation. These findings support the sustainable production of agricultural products by providing actionable strategies, such as the removal of weeds to eliminate habitats for vectors like whiteflies and aphids and the targeted management of high-incidence weeds from the Asteraceae, Solanaceae, and Oxalidaceae families to prevent and control the spread of EuLCV.

## 1. Introduction

*Passiflora edulis* Sims, commonly known as passion fruit, is a perennial vine belonging to the family Passifloraceae, native to South America, including Brazil [1]. Two subspecies of passion fruit are known, *P. edulis* and *P. edulis* f. *flavicarpa*; the former bears fruits that turn dark purple in color when ripe, whereas the latter bears bright-yellow fruits [2]. The global production of passion fruit amounts to approximately 1400 million kg/year, with Brazil being the largest producer, generating approximately 600 million kg/year [3]. Passion fruit is widely cultivated in tropical and subtropical regions as an attractive high-value crop, with South America continuing to be the main producer [4].

*P. edulis* was introduced in the Republic of Korea in 1989, and with the beginning of its commercial cultivation in greenhouses in 2012, the species has been widely cultivated nationwide [5,6]. Due to the impact of global warming from climate change on agricultural practice in the Republic of Korean Peninsula, subtropical crop farming in the Republic of Korea expanded in terms of both species diversity and area of cultivation, starting from the southern region of Korea: the area dedicated to subtropical crop farming expanded from 102.4 ha to 171.3 ha, supporting nine and ten crop types in 2017 and 2021, respectively [7].

The cultivation and production of passion fruit are affected by various pathogens, such as viruses, bacteria, and fungi. Particularly, diseases caused by viruses pose extremely severe constraints for passion fruit, ultimately reducing crop yield. When passion fruit is infected with a virus, the yield and quality of the fruits are continuously affected during the cultivation period, ultimately leading to crop failure and substantial financial losses for farms [8]. Passion fruit can be virally infected through various routes and is susceptible to infection by more than 30 different viruses globally [8]. These include members of the genus *Potyvirus*, such as East Asian Passiflora virus (EAPV) [9], Telosma mosaic virus (TeMV) [10], and passion fruit woodiness virus (PWV) [11]; the genus *Carlavirus*, such as Passiflora latent virus (PLV) [12]; the genus *Begomovirus*, such as passion fruit severe leaf distortion virus (PSLDV) [13], Euphorbia leaf curl virus (EuLCV), papaya leaf curl Guangdong virus (PaLCuGdV) [14], and papaya leaf curl China virus (PaLCuCNV) [15]; and the genus *Cytorhabdovirus*, such as citrus-associated rhabdovirus (CiaRV) [16]. Particularly, Huang et al. [17] reported that TeMV, EAPV, CiaRV, and PLV were detected in the majority of *P. edulis* production areas in China.

Based on the report of Hong [18], among more than 30 species of viruses, cucumber mosaic virus (CMV), EAPV, EuLCV, PaLCuGdV, and tomato yellow leaf curl virus (TYLCV) were detected in the *P. edulis* cultivation sites in the Republic of Korea; particularly, *P. edulis* infections by EuLCV, PaLCuGdV, and EAPV were first detected in the Republic of Korea. Additionally, CMV comprises an extremely broad host range [19] and uses vector transmission by aphids to infect *P. edulis* [20]. Additionally, in the study by Jeon et al. [5] on the incidence of viral infection by six viruses, infection with TYLCV was not detected, including PWV in *P. edulis* cultivated in the Republic of Korea. Therefore, it was concluded that the main pathogens accounting for the viral diseases of *P. edulis* in the Republic of Korea are CMV, EAPV, PaLCuGdV, and EuLCV. These viruses are mainly transmitted through both vertical transmission via cuttings of infected seedlings and horizontal transmission by insect vectors or mechanical contact. Particularly, insect vectors are known to spread using plants of the Solanaceae family as hosts [21]. The symptoms of viral infection include chlorotic mottle, yellow mosaic, and curling of leaves, and for fruits, the infection causes woodiness of the pericarp and malformation, leading to a substantial impact on the yield and quality of fruits [22].

Some of the examples representing the recent trends of research on viral diseases of passion fruit are as follows: the first report of PLV infection of *P. edulis* in the Republic of Korea and China [23,24], the first report of passion fruit green spot virus (PfGSV) infection of yellow passion fruit (*P. edulis* f. *flavicarpa*) in Columbia [25], a study on the occurrence and distribution of major viruses infecting *P. edulis* [26], and a study on simultaneous detection and differentiation of four viruses in *P. edulis* plants using multiplex RT-PCR [17]. Most earlier studies reported the first outbreak of the viruses, diagnostic methods, and distribution of incidences of viral infection. However, while inoculation experiments were performed to identify potential host plants for the Cowpea aphid-borne mosaic virus affecting yellow passion fruit orchards [27], studies investigating the relationship between viral infection of P. edulis and that of weeds growing around *P. edulis* cultivation sites are limited. This information could be useful for preventing the infection and spread of viral diseases in *P. edulis* cultivation areas.

Therefore, this study investigated the types of viruses detected in major *P. edulis* cultivation sites in the Republic of Korea and the patterns of viral infection in *P. edulis* to support sustainable agriculture. Additionally, the trend of viral spread was examined by analyzing the correlation between viral infections in *P. edulis* and virus-infected weeds surrounding the cultivation sites, which may serve as potential sources of viral infection for *P. edulis*. The study objective was to enhance understanding of the latest trends in viral infections affecting *P. edulis* in Republic of Korean cultivation sites. This information is crucial not only for preventing the spread of viral diseases but also for implementing effective biological control measures, thereby supporting the sustainable production of agricultural products and enhancing the resilience of agricultural systems.

## 2. Materials and Methods

### 2.1. Selection of Study Area and P. edulis Sample Collection

Initially, to analyze the current status of viral infection of *P. edulis*, the main cultivation sites of *P. edulis* in Jeonbuk State, the Republic of Korea, were monitored. As of 2021, 10 species of subtropical crops were cultivated over an area of 171.3 ha in the Republic of Korea, of which *P. edulis* was the second most cultivated crop after *Mangifera indica* (mango), taking up 34.8 ha in Jeonbuk State, which is the second-largest producer of *P. edulis* in the Republic of Korea, with *P. edulis* accounting for 9.14 ha out of 13.3 ha of the area for cultivation of subtropical crops. In Jeonbuk State, the main cultivation sites of *P. edulis* are located in Gimje, Namwon, Iksan, and Wanju, and the plant has emerged as a new cash crop [7]. Therefore, the study area was selected to encompass diverse locations within Jeonbuk State, specifically Gimje (2 greenhouses), Namwon (3 greenhouses), Iksan (1 greenhouse), and Wanju (2 greenhouses), where many *P. edulis* farms are situated (Figure 1).

The study spanned five years (2017–2021), and samples were collected from 10 to 15 *P. edulis* plants showing symptoms of viral infection per visit at each farm three times a year. To analyze viral infection, three leaves were collected from each plant. Due to differences in cultivation conditions and scale among the *P. edulis* farms, the number of plants sampled was inconsistent.

### 2.2. PCR/RT-PCR Assays for Detection and Characterization of Viral Infection Status of P. edulis

For the detection and analyses of viral infection in *P. edulis* samples, a total of 283 samples were collected and analyzed using polymerase chain reaction (PCR) and reverse transcription PCR (RT-PCR) assays. For the extraction of viral DNA/RNA, 0.1 g of the collected leaves was placed in a 1.5 mL Eppendorf tube, ground with liquid nitrogen, and total DNA/RNA was extracted using the Viral Gene-spin Viral DNA/RNA Extraction Kit (iNtRON Biotechnology, Seongnam, Republic of Korea). The purity and concentration of the extracted DNA/RNA were determined using a NanoDrop spectrophotometer (Thermo Fisher Scientific, Seoul, Republic of Korea), and the extracted nucleic acids were divided into small volumes and stored at −80 °C for use in future experiments. Virus-specific primers developed by Hong [18] for the diagnosis of five major *P. edulis* viruses in the Republic of Korea (EuLCV, PaLCuGdV, TYLCV, CMV, and EAPV) were used for detection and diagnosis of infection by these viruses (Table 1).

The PCR assay for diagnosis of viral infection from DNA viruses (EuLCV, PaLCuGdV, and TYLCV) was performed using GoTaq DNA Polymerase (Promega, Seoul, Republic of Korea), 2.5 mM dNTP, 5X Taq polymerase buffer, and 25 mM MgCl_2_. DNA (10 ng) was extracted from *P. edulis* leaves and leaves of the surrounding weeds, and 10 pmol of each virus-specific primer set was added. The SimpliAmp Thermal Cycler (ABI, Thermo Fisher Scientific, Seoul, Republic of Korea) was used for the reaction, starting with an initial denaturation at 95 °C for 3 min, followed by 35 cycles at 94 °C, 55 °C, and 72 °C for 20 s, 30 s, and 1 min, respectively, and a final extension at 72 °C for 10 min.

The one-step RT-PCR for diagnosis of infection by RNA viruses (CMV and EAPV) was performed using SuPrimescript RT-PCR Premix (Genetbio, Daejeon, Republic of Korea); 10 ng of the extracted RNA and 10 pmol of the primer set were added, and the assay was performed according to the manufacturer’s protocol. The reaction conditions for RT-PCR were as follows: reverse transcription reaction at 50 °C for 30 min and 95 °C for 10 min, followed by 35 cycles at 95 °C, 55 °C, and 72 °C for 30 s, 30 s, and 1 min, respectively, and 72 °C for 5 min to complete the reaction.

The size of the PCR and RT-PCR amplicons was determined using electrophoresis at 100 V for 30 min in a 1.2% agarose gel added with 1 × TBE buffer and DNA stain (GoodView, Bratislava, Slovak Republic), including 100 bp DNA Ladder (Invitrogen, Carlsbad, CA, USA). Then, WSE-5300 Printgraph CMOS I (Atto, Tokyo, Japan), a gel imaging system, was used for visualization of the infection status.

The infection rate (%) was calculated using the formula in Odum [28] (Equation (1)). The infection rate was calculated using the state of infection determined using PCR and RT-PCR assays for the collected samples.(1)$$Infection\;rate\;\left(\%\right)=\frac{No.\;of\;samples\;confirmed\;with\;infection\;(plants)}{No.\;of\;total\;samples\;(plants)}\times 100$$

### 2.3. Analysis of Viral Infection Status of Weeds Surrounding the P. edulis Cultivation Sites and Infection Status by Weed Species

This study built on the investigations conducted in 2017 which identified a high rate of viral infection in *P. edulis* cultivation sites in the Republic of Korea. Consequently, the status of viral infections in weeds, including both herbaceous and woody seedlings, surrounding these cultivation sites was investigated to examine the relationship between viral infections in *P. edulis* and in the adjacent weeds. Weed surveys targeted all types of weeds surrounding *P. edulis* cultivation sites from 2018 to 2021. Each farm was visited three times a year, and during each visit, 10–15 different species of weeds were collected. Since the weeds showed no apparent symptoms of viral infection, we selected and collected various species, avoiding duplicates as much as possible. Over these four years, a total of 461 samples were collected to analyze the major viral infection status and types of infection (single or mixed viral infection). For testing of viral infection of weeds, five viruses were used, just as in the analysis of viral infection of *P. edulis*. For the diagnosis of viral infection based on PCR or RT-PCR for weeds, the same primers for diagnosis were used as those in the testing of *P. edulis* viruses. To investigate whether weeds serve as a source of primary inoculum that could affect the viral infection of *P. edulis*, the viral status of various weed families was analyzed. Infected weeds were classified by family, and their potential role in transmitting viruses to *P. edulis* was assessed based on these classifications. For the classification of the weeds, the Republic of Korea Biodiversity Information System (http://www.nature.go.kr (accessed on 2 February 2025) and the *Handbook of Useful Weeds* [29] were used for identification.

### 2.4. Analysis of Correlation Between the Viruses Identified in P. edulis Cultivation Sites and the Infecting Weeds Surrounding the Sites

To examine the relationship between the viruses of *P. edulis* cultivation sites and the infecting weeds surrounding the sites, data from the 4-year monitoring period (2018–2021) on the viruses infecting weeds were utilized to analyze the frequency of virus infection in the *P. edulis* cultivation sites and on the viral infection of weeds surrounding the sites to analyze the correlation between these viruses. For correlation analysis, a non-parametric correlation analysis (Spearmen’s rank correlation test) was performed using the number of samples with viral infection according to the virus types infecting *P. edulis* and weeds for each year in the study period. IMB SPSS Statistics Ver. 21.0 software (IBM Co., Armonk, NY, USA) was used for analysis, and statistical significance was set at α = 0.05.

## 3. Results and Discussion

### 3.1. Status of Viral Infection of P. edulis

Analysis of the total number of 283 *P. edulis* samples for virus infection status showed that PaLCuGdV and EuLCV had high infection rates of 74.6% and 62.5%, respectively, followed by EAPV and CMV at infection rates of 40.3% and 22.6%, respectively. Contrastingly, infection with TYLCV was not detected in *P. edulis* cultivation sites during the study period. EAPV consistently showed a high infection rate of over 30%, except during the year 2021. The rate of infection with CMV continued to increase by more than 10% from the third year (2019) in the study period, reaching 50% in 2021 (Table 2).

In *P. edulis* cultivation sites in the Republic of Korea, PaLCuGdV, EuLCV, and EAPV were the main virus types with reports of high incidences. These viruses are classified into the genera *Begomovirus* (PaLCuGdV and EuLCV) and *Potyvirus* (EAPV), indicating the variety of virus types affecting *P. edulis* in the Republic of Korea. Particularly, Hong [18] reported that in *P. edulis* cultivation sites in the Republic of Korea over a 3-year period (2014–2016), PaLCuGdV, EuLCV, EAPV, and CMV showed infection rates of 78.5%, 69.6%, 34.0%, and 13.8%, respectively; the ranking of the infection percentages of these viruses aligns with the findings of the current study. However, TYLCV infection (9.8%) was confirmed in the report by Hong [18], whereas TYLCV infection was not confirmed in this study. Additionally, the research conducted by Jeon et al. [5] on the incidence of viral diseases of *P. edulis* in the Republic of Korea in 2020 also reported no incidence of TYLCV infection.

PaLCuGdV and EuLCV of the genus *Begomovirus* are pathogens of viral diseases, characterized by symptoms such as leaf curling and mottling. In Asia, these viruses were first reported in a passion fruit cultivar named ‘Tainung No. 1’ of Taiwan [14]. In the Republic of Korea, infection with PaLCuGdV was confirmed in a bell pepper farming greenhouse around one of the *P. edulis* cultivation sites, and it was the first case of the virus PaLCuGdV, introduced along with the introduction of *P. edulis* and spreading to the bell pepper, a crop growing in the neighboring site of a *P. edulis* cultivation site [30]. The dominance of PaLCuGdV and EuLCV in the Republic of Korea can be attributed to the introduction of *P. edulis*, one of the subtropical crops, into the Republic of Korea following climate change and the use of infected cuttings considering the characteristics of the genus *Begomovirus* related to mechanical transmission [31].

Regarding the incidence of EAPV of the genus *Potyvirus*, the first case of *P. edulis* infection was reported in *P. edulis* in Jeollanam-do, the Republic of Korea, in 2014. EAPV infection was also confirmed in *P. edulis* grown in Jeju Island in 2015 [32]. The infection rate of EAPV in *P. edulis* was in the range of 19.2–75.0% (average: 40.3%) (Table 2), indicating that EAPV should also be treated as one of the major viruses causing viral diseases of *P. edulis*. For a long time, EAPV was classified as PWV, although it was identified and reclassified as a new species through studies in Japan [9] and Taiwan [33] through host plant assays and phylogenetic analysis using coat proteins and full-genome analysis.

CMV of the genus *Cucumovirus* has a wide host range of over 1000 species of 85 families [34]. Particularly, CMV is widely distributed globally and known to be the most important virus affecting vegetables and ornamentals [35]. Here, the infection rate of CMV infection in *P. edulis* was low at the initial stage of cultivation, although it increased over time with every year of cultivation (Table 2). The differences in the trend of infection rate are attributable to the characteristics of CMV, which is endemic and has a wide host range, unlike the trend of decreasing infection rate of EAPV, which is exotic and has a narrow host range. CMV infection of the yellow passion fruit (*P. edulis* f. *flavicarpa*) in Brazil shows symptoms of bright-yellow mottling only for some leaves of the crop [36]. Furthermore, new leaves did not exhibit any symptoms of viral infection, and no viruses were detected even after 40 to 85 days from seedling inoculation [36]. Although the symptoms of CMV infection were similar to those in purple passion fruit (*P. edulis*) in the Republic of Korea, the infection rate of CMV is increasing, indicating the necessity for accurate analysis of the damage caused by the viral infection and provision of useful information that can be applied in actual cultivation sites.

Analysis of the status of virus infection in *P. edulis* revealed that 245 samples, or 86.6%, of the 283 samples were infected. Analyzing the type of viral infection of *P. edulis*, mixed infection accounted for 65.4% and single infection accounted for 21.2%, with the percentage of mixed infection being more than three times that of single infection (Table 3). Mixed infection by multiple viruses is known to be a common phenomenon, occurring naturally [37]. When mixed infections occur in plants, the severity of the disease and symptoms may be exacerbated as a result of synergistic interactions between different viruses, depending on the virus titers and the mobility of the viruses [31]. It has been determined that mixed infection may cause more devastating damage to the yield or quality of *P. edulis* than the damage from a single infection [17]. Therefore, it was deemed necessary to determine the exact extent of the damage caused by mixed infection by virus type in the *P. edulis* cultivation sites. Additionally, further research is required to investigate the methods of precise diagnosis for mixed viral infections of *P. edulis*.

### 3.2. Status of Viral Infection of Weeds Surrounding the P. edulis Cultivation Sites

An analysis of viral infection status in 461 weed samples surrounding *P. edulis* cultivation sites revealed four types of virus infections in the case of *P. edulis*. Furthermore, the order of infection rates of these main viruses was similar to that of P. edulis. PaLCuGdV and EuLCV showed high infection rates of 12.4% and 11.3%, respectively, followed by CMV and EAPV at infection rates of 8.2% and 0.2%, respectively. Additionally, no infection with TYLCV was confirmed in weeds surrounding the *P. edulis* cultivation sites. In weeds, EAPV was detected in only one sample and showed a lower infection rate than CMV (Table 4).

Generally, EAPV was reported to be dispersed through infected seedlings (plants) and has no wild host [38]. Additionally, three species of aphids (*Aphis gossypii*, *Myzus persicae*, and *Hyperomyzuz lactucae*) carry EAPV as vectors in a non-persistent manner, although no aphid clones were observed in *P. edulis* [39]. Here, a few cotton aphids (*Aphis gossypii*) were identified on some young leaves in the *P. edulis* cultivation sites; however, due to the lack of reproduction or colony formation by the aphids, it would have been difficult to spread the virus to weeds.

Analysis of the current status of virus infection in weeds surrounding the *P. edulis* cultivation sites revealed that 101 samples, or 21.9%, of the 461 samples were infected. Regarding the infection type in weeds surrounding *P. edulis* cultivation sites, the rates of single and mixed infections were 12.8% and 9.1%, respectively, indicating that the rate of single infection was higher by about 1.4 times than that of mixed infection (Table 5), showing the opposite trend compared with that of the infection type in *P. edulis* (see Table 3). In weeds, many cases existed of mixed infection of PaLCuGdV and EuLCV, consistent with the trend in *P. edulis* (Table 6). Furthermore, the correlation analyses conducted on these two viruses revealed a statistically significant positive correlation (Table 7). Therefore, further comprehensive research is required to explore the various host ranges of PaLCuGdV and EuLCV (which are the viruses that were introduced into the Republic of Korea along with the introduction of *P. edulis*), the extent of damage caused by mixed infection of these two viruses, and the spread of the viruses to crops of neighboring sites.

### 3.3. Current Status of Viral Infection of Weeds Surrounding the P. edulis Cultivation Sites by Weed Species

A total of 461 weed samples showed differences in terms of frequency of infection by viruses depending on the type of weed. The weeds infected with the four types of viruses included in this study were classified into 27 families and 56 species (excluding one unidentified species of weed). The family of weeds with the highest frequency of viral infection was Asteraceae, and among the four viruses, PaLCuGdV was dominant. From the details on species of weeds, four species from the weeds of Amaranthaceae were infected, and three species were infected from weeds of Poaceae, Lamiaceae, and Fabaceae. Of the four families, infection with PaLCuGdV was dominant for weeds of the Amaranthaceae and Fabaceae families, and weeds of Poaceae were mainly detected with CMV infection (see Table 6). It was reasoned that this was associated with the fact that CMV shows a high infection rate in plants of the Asteraceae and Poaceae families in cultivation sites of upland crops in the Republic of Korea [40].

The total number of detected viral infections in the classified weeds (excluding one unidentified species of weed) was 146, of which PaLCuGdV showed the highest incidence (38.4%), followed by EuLCV (34.9%), CMV (26.0%), and EAPV (0.7%). Of the four types of viruses infecting weeds, EAPV infection was only detected in *Prunus serrulate* f. *spontanea* (Rosaceae) in 2018 (see Table 4 and Table 6), and after 2018, no infection was detected in the weeds surrounding the *P. edulis* cultivation sites. Regarding viruses of the genus *Potyvirus*, such as EAPV, the host plants only included those belonging to Fabaceae, Solanaceae, Chenopodiaceae, and Amaranthaceae [21]. An earlier study on the host range of EAPV [41] also reported that plants of the Solanaceae and Amaranthaceae families showed susceptibility to this virus, whereas no infection was detected in *Vigna unguiculata* or *Phaseolus vulgaris* var. *humilis* of the Fabaceae family. Here, EAPV was detected in *P. serrulate* f. *spontanea*, and considering that Plum pox virus, a virus infecting Rosaceae stone fruits (*Prunus* spp.) such as *P. persica* (peaches) and *P. salicina* (plums), also belongs to the genus *Potyvirus* [42], it was considered that pathogenicity testing would be needed for EAPV infection of stone fruits (*Prunus* spp.).

The infection rate of EuLCV was the highest (47.1%) in *Solanum nigrum*, a weed belonging to the Solanaceae family and *Capsicum annuum* (chili peppers), a crop host (see Table 6). As *C. annuum* is commonly found surrounding *P. edulis* cultivation sites in the Republic of Korea and is a commercially valuable crop [43], concerns exist about the spread of viral infection and an accurate analysis of the damage from the viral infection is considered necessary. Furthermore, natural occurrence of EuLCV infection was reported in *Carica papaya* (papaya) grown in the same site as *P. edulis* infected with EuLCV at a farm growing subtropical crops in Haenam, the Republic of Korea [44]. Therefore, it is imperative to conduct further research on viral infections associated with unidentified host plants, and caution should be exercised in the introduction and cultivation of new crops.

### 3.4. Correlation Between Viruses Infecting P. edulis in the Cultivation Sites and Infecting Weeds Surrounding the Cultivation Sites

Analysis of the correlation between viruses infecting *P. edulis* and infecting weeds surrounding the cultivation sites revealed no correlation between the same viruses (see Table 7). The role of weeds in the transmission of viruses and their significant impact on the viral infection of other plants has long been recognized. Generally, weeds play an important role in terms of virus overwintering and as a source of insect-borne viral diseases [45,46]. However, here, no correlation between the same viruses was identified between weeds and *P. edulis*. In the Republic of Korea, *P. edulis* was grown year-round in greenhouse facilities rather than outdoors during the cultivation period of 4–5 years. Therefore, the viruses infecting weeds impact inter-virus relationships in mixed infection in the same group rather than the transmission of viruses to *P. edulis*.

Another point to consider is that the spread of *P. edulis* virus in the greenhouse started from the row near the side-wall window. This pattern was consistent with that of the findings of Ko et al. [47], who reported that *C. annuum* cultivated in a greenhouse showed a high infection rate with tomato spotted wilt virus (TSWV) in rows adjacent to the side-wall window and that the infection rate decreased with increasing distance from the window, demonstrating that the virus was introduced from the area outside the side-wall window.

Analysis of the correlation between viruses in the same group revealed that PaLCuGdV showed a significant positive correlation with EuLCV (ρ = 0.57 **) and CMV (ρ = 0.23 **) and that EuLCV showed a significant positive correlation with EAPV (ρ = 0.33 **). For weeds surrounding the *P. edulis* cultivation sites, a similar pattern of correlation was identified as in the case of *P. edulis*, i.e., PaLCuGdV showed a significant positive correlation with EuLCV (ρ = 0.61 **), and EuLCV showed a significant positive correlation with CMV (ρ = 0.14 *) and EAPV (ρ = 0.15 *) (see Table 7).

In *P. edulis*, the infection rates of PaLCuGdV, EuLCV, EAPV, and CMV were 74.6%, 62.5%, 40.3%, and 22.6%, respectively (see Table 2). For weeds, the infection rates were 12.4%, 11.3%, 0.2%, and 8.2%, respectively (see Table 4). Among the four viruses, PaLCuGdV showed the highest incidence in *P. edulis* and weeds. However, in the results of the correlation analysis of viruses infecting *P. edulis* and weeds, EuLCV showed a positive correlation with all the other viruses, i.e., PaLCuGdV, EAPV, and CMV (see Table 7). Specifically, in the case of viral infection in *P. edulis* cultivation sites of the Republic of Korea, EuLCV infection has an impact on mixed infection with other viruses. Hong [18] reported that the rate of mixed infection of EuLCV and PaLCuGdV was the highest at 28.9% in a greenhouse cultivation site of *P. edulis*. A study by Jeon et al. [5] on the incidence of viral disease on purple passion fruit (*P. edulis*) reported that the rate of single infection was 25%, whereas the rate of multiple viral infections was as high as 75%. Furthermore, the incidence of mixed infection of EuLCV, PaLCuGdV, and CMV was high in the region of Jeonbuk State [5]. When the incidence of EuLCV was first reported in Taiwan, only mixed infection of EuLCV with PaLCuGdV was detected in *P. edulis* cultivation sites, and no single infection of EuLCV was detected. Additionally, in the artificial inoculation tests, all plants receiving multiple inoculations exhibited mixed infections [14].

Mixed viral infection of *P. edulis* is generally known to be a factor causing severe symptoms and further damage to the yield and commercial value of the fruit [17]. In *P. edulis*, mixed infection with EuLCV, PaLCuGdV, and EAPV resulted in more severe foliar mosaic symptoms and increased malformation of fruits [33], i.e., in *P. edulis* cultivation, EuLCV increases damage when it occurs as a mixed infection rather than as a single infection. Thus, for *P. edulis* grown in greenhouses for 4–5 years, it is crucial to eradicate weeds surrounding the cultivation area and outside the side-wall windows during the fallow period before replacement of the rooted cuttings to eliminate the habitats of insect vectors (whiteflies and aphids). Particularly, regarding strategies to prevent infection and spread of EuLCV, focusing on the control of weeds of the Asteraceae (11 incidences), Solanaceae (8 incidences), and Oxalidaceae (5 incidences) families would be effective in preventing mixed infection by EuLCV (see Table 6).

## 4. Conclusions

In the Republic of Korea, the cultivation of subtropical crops, including *P. edulis*, has been increasing as a result of global warming. Despite its commercial cultivation starting in 2012 and its ranking as the second-largest subtropical crop after mango, *P. edulis* faces challenges due to viral infections affecting its growth, yield, and quality. This study aimed to address these challenges by examining the types of viruses infecting *P. edulis* and surrounding weeds, assessing the current status of viral infections, the nature of these infections (single or mixed), and their correlations with transmission dynamics.

The findings of the study revealed that both *P. edulis* and nearby weeds were infected with various viruses, such as PaLCuGdV, EuLCV, CMV, and EAPV, with PaLCuGdV showing the highest infection rate. While no direct correlation was found between the presence of the same viruses in *P. edulis* and weeds, indicating potential interactions among different viruses, the study highlighted that EuLCV infection could exacerbate symptoms when coinfected by other viruses. This suggests that strategically focused management of specific weeds, particularly from the Asteraceae, Solanaceae, and Oxalidaceae families, is essential to prevent the spread of EuLCV.

Furthermore, while surrounding weeds can significantly impact viral infections and spread [48], this study did not find a significant correlation regarding the transmission of the same virus between weeds and *P. edulis*. This underscores the importance of implementing preventive measures within greenhouses, as virus transmission seems to occur mainly internally rather than externally.

The original contribution of this study lies in its comparison of viruses infecting *P. edulis* with those affecting surrounding weeds at cultivation sites. These findings provide valuable insights for the strategic management of viral diseases in *P. edulis* cultivation, supporting sustainable agricultural production. However, given the regional focus of the study, further nationwide investigation and analysis are necessary to enhance result reliability. Such extended research will aid in identifying regional differences and equip *P. edulis* growers with effective strategies for viral infection prevention and control. Additionally, further research on the impact of cultivation environments on viral infections and *P. edulis* productivity will enhance our understanding and lead to more effective, sustainable virus control strategies in *P. edulis* cultivation, ultimately supporting the long-term viability of subtropical agriculture in the Republic of Korea.

## Figures and Tables

**Figure 1 viruses-17-00383-f001:**
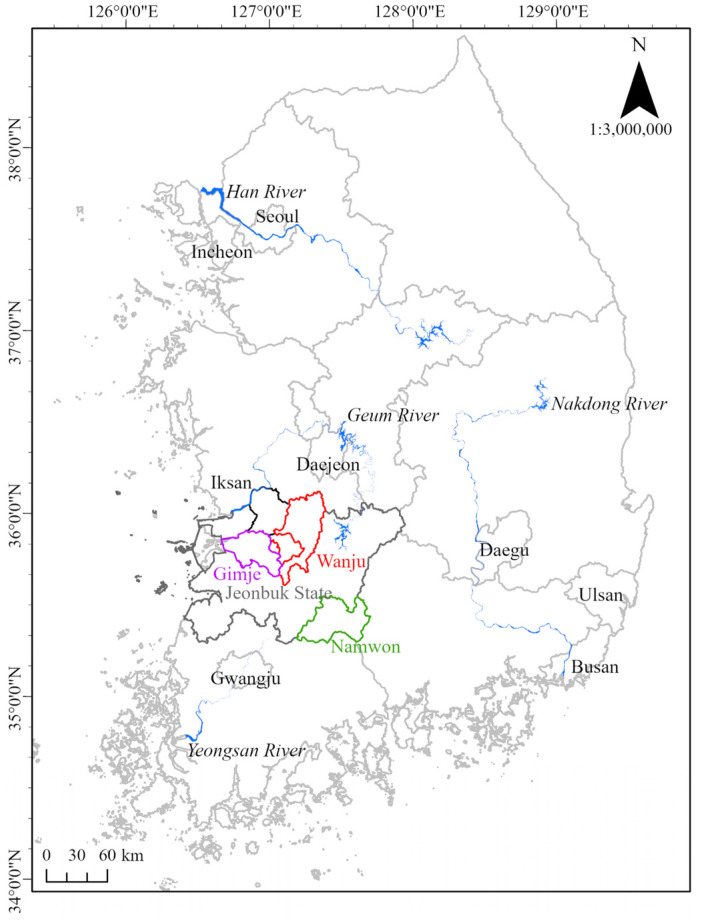
Map of research area.

**Table 1 viruses-17-00383-t001:** List of viruses and their specific diagnostic primers used in this experiment.

Virus	Primer	Primer Sequences (5′→3′)	Size (bp)	Accession No. Referred
EuLCV	EuLCV-F	AGTGGTCCCCCCTCCACTAAC	339	AJ811911.1
	EuLCV-R	CAGCCTCCGTCGAACCTTCG		
PaLCuGdV	PaLCuGdV-m-2F	CTGTCTTACGTGCAAGGA	605	MZ130299.1
	PaLCuGdV-m-2R	GCTTGCATATTGACCACCAG		
TYLCV	TYLCV 1F	GTC AAC CAA TCA AAT TGC ATC CTC AA	712	AM691759.1
	TYLCV 1R	GTC CAA AAT CCA TTG GGC		
CMV	CMV 1-5 u	CTTGTGCGTTTRATGGCTACGAAGGC	473	M57602.1
	CMV 1-5 d	CACGGACCGAAGTCCTTCCGAAGAAA		
EAPV	EAPV-IB-1F	CATTGATAATGGCACCTCACC	231	MH922997.1
	EAPV-IB-1R	AGCCAAACTCAAGTCCCTCA		

**Table 2 viruses-17-00383-t002:** Status of viral infection of *P. edulis*.

Year	PaLCuGdV	EuLCV	EAPV	CMV	TYLCV	Uninfected	No. of Samples Tested
2017	28 (87.5)	28 (87.5)	24 (75.0)	1 (3.1)	0 (0.0)	1	32 (100.0)
2018	33 (58.9)	30 (53.6)	22 (39.3)	6 (10.7)	0 (0.0)	12	56 (100.0)
2019	62 (73.8)	37 (44.0)	26 (31.0)	19 (22.6)	0 (0.0)	11	84 (100.0)
2020	67 (78.8)	64 (75.3)	37 (43.5)	25 (29.4)	0 (0.0)	11	85 (100.0)
2021	21 (80.8)	18 (69.2)	5 (19.2)	13 (50.0)	0 (0.0)	3	26 (100.0)
Total	211 (74.6)	177 (62.5)	114 (40.3)	64 (22.6)	0 (0.0)	38	283 (100.0)

Values in parentheses represent the infection rates (%). For mixed virus infections, each virus was counted individually.

**Table 3 viruses-17-00383-t003:** Viral infection types of *P. edulis*.

Year	Infection Type			No. of Samples Tested
	Single	Mixed	Total	
2017	1 (3.1)	30 (93.8)	31 (96.9)	32 (100.0)
2018	14 (25.0)	30 (53.6)	44 (78.6)	56 (100.0)
2019	32 (38.1)	41 (48.8)	73 (86.9)	84 (100.0)
2020	11 (12.9)	63 (74.1)	74 (87.1)	85 (100.0)
2021	2 (7.7)	21 (80.8)	23 (88.5)	26 (100.0)
Total	60 (21.2)	185 (65.4)	245 (86.6)	283 (100.0)

Values in parentheses represent the infection rates (%).

**Table 4 viruses-17-00383-t004:** Status of viral infection of weeds surrounding the *P. edulis* cultivation sites.

Year	PaLCuGdV	EuLCV	EAPV	CMV	TYLCV	Uninfected	No. of Samples Tested
2018	12 (48.0)	10 (40.0)	1 (4.0)	0 (0.0)	0 (0.0)	12	25 (100.0)
2019	6 (9.2)	2 (3.1)	0 (0.0)	5 (7.7)	0 (0.0)	54	65 (100.0)
2020	30 (22.7)	29 (22.0)	0 (0.0)	11 (8.3)	0 (0.0)	85	132 (100.0)
2021	9 (3.8)	11 (4.6)	0 (0.0)	22 (9.2)	0 (0.0)	209	239 (100.0)
Total	57 (12.4)	52 (11.3)	1 (0.2)	38 (8.2)	0 (0.0)	360	461 (100.0)

Values in parentheses represent the infection rates (%). For mixed virus infections, each virus was counted individually.

**Table 5 viruses-17-00383-t005:** Infection types of weeds surrounding the *P. edulis* cultivation sites.

Year	Infection Type			No. of Samples Tested
	Single	Mixed	Total	
2018	4 (16.0)	9 (36.0)	13 (52.0)	25 (100.0)
2019	10 (15.4)	1 (1.5)	11 (16.9)	65 (100.0)
2020	26 (19.7)	21 (15.9)	47 (35.6)	132 (100.0)
2021	19 (7.9)	11 (4.6)	30 (12.6)	239 (100.0)
Total	59 (12.8)	42 (9.1)	101 (21.9)	461 (100.0)

Values in parentheses represent the infection rates (%).

**Table 6 viruses-17-00383-t006:** Status of viral infection of weeds surrounding the *P. edulis* cultivation sites according to weed species.

Family	Species (Tested Plants)	No. of Viral Infections				
		PaLCuGdV	EuLCV	EAPV	CMV	Total
Asteraceae (14 species)	*Erigeron annuus* (14)	3	3	0	0	6
	*Sonchus asper* (8)	2	1	0	2	5
	*Erigeron canadensis* (19)	3	0	0	0	3
	*Taraxacum mongolicum* (12)	2	0	0	1	3
	*Bidens frondosa* (5)	1	1	0	0	2
	*Youngia japonica* (1)	1	1	0	0	2
	*Cirsium japonicum* var. *maackii* (1)	1	1	0	0	2
	*Eclipta thermalis* (17)	1	1	0	0	2
	*Artemisia annua* (1)	1	0	0	1	2
	*Sonchus oleraceus* (9)	0	1	0	0	1
	*Artemisia indica* (3)	0	1	0	0	1
	*Centipeda minima* (14)	0	1	0	0	1
	*Erigeron sumatrensis* (1)	0	0	0	1	1
	*Symphyotrichum expansum* (2)	0	0	0	1	1
	Subtotal (107)	15	11	0	6	32
		(46.9)	(34.4)	(0.0)	(18.8)	(100.0)
Amaranthaceae (4 species)	*Amaranthus blitum* (4)	2	1	0	0	3
	*Amaranthus tricolor* (2)	1	1	0	0	2
	*Achyranthes bidentata* var. *japonica* (10)	1	0	0	1	2
	*Chenopodium album* var. *centrorubrum* (14)	2	0	0	1	3
	Subtotal (30)	6	2	0	2	10
		(60.0)	(20.0)	(0.0)	(20.0)	(100.0)
Poaceae (3 species)	*Digitaria ciliaris* (16)	3	3	0	3	9
	*Setaria viridis* (1)	0	0	0	1	1
	*Poa annua* (7)	0	0	0	1	1
	Subtotal (24)	3	3	0	5	11
		(27.3)	(27.3)	(0.0)	(45.5)	(100.0)
Lamiaceae (3 species)	*Lamium amplexicaule* (3)	1	1	0	1	3
	*Salvia rosmarinus* (1)	1	1	0	0	2
	*Lavandula angustifolia* (5)	0	0	0	1	1
	Subtotal (9)	2	2	0	2	6
		(33.3)	(33.3)	(0.0)	(33.3)	(100.0)
Fabaceae (3 species)	*Trifolium repens* (3)	1	1	0	0	2
	*Glycine max* subsp. *soja* (3)	1	0	0	0	1
	*Amorpha fruticosa* (1)	0	0	0	1	1
	Subtotal (7)	2	1	0	1	4
		(50.0)	(25.0)	(0.0)	(25.0)	(100.0)
Solanaceae (2 species)	*Solanum nigrum* (19)	4	5	0	2	11
	*Capsicum annuum* (4)	1	3	0	2	6
	Subtotal (23)	5	8	0	4	17
		(29.4)	(47.1)	(0.0)	(23.5)	(100.0)
Liliaceae (2 species)	*Chlorophytum comosum* var. *variegatum* (3)	1	2	0	1	4
	*Liriope muscari* (4)	1	1	0	0	2
	Subtotal (7)	2	3	0	1	6
Caryophyllaceae (2 species)	*Stellaria aquatica* (4)	2	1	0	0	3
	*Stellaria media* (3)	1	0	0	1	2
	Subtotal (7)	3	1	0	1	5
Ranunculaceae (2 species)	*Aquilegia buergeriana* var. *oxysepala* (1)	1	1	0	0	2
	*Ranunculus sceleratus* (1)	1	0	0	0	1
	Subtotal (2)	2	1	0	0	3
Boraginaceae (2 species)	*Trigonotis peduncularis* (1)	1	1	0	0	2
	*Bothriospermum zeylanicum* (1)	0	0	0	1	1
	Subtotal (2)	1	1	0	1	3
Brassicaceae (2 species)	*Cardamine flexuosa* (3)	0	1	0	1	2
	*Turritis glabra* (1)	0	1	0	0	1
	Subtotal (4)	0	2	0	1	3
Euphorbiaceae (2 species)	*Acalypha australis* (12)	1	0	0	0	1
	*Euphorbia maculata* (1)	0	0	0	1	1
	Subtotal (13)	1	0	0	1	2
Oxalidaceae (1 species)	*Oxalis corniculate* (49)	7 (41.2)	5 (29.4)	0 (0.0)	5 (29.4)	17 (100.0)
Plantaginaceae (1 species)	*Plantago asiatica* (3)	1	3	0	0	4
Rosaceae (1 species)	*Prunus serrulate* f. *spontanea* (1)	1	1	1	0	3
Apiaceae (1 species)	*Peucedanum japonicum* (4)	1	1	0	1	3
Cyperaceae (1 species)	*Cyperus amuricus* (1)	1	1	0	0	2
Cannabaceae (1 species)	*Humulus scandens* (2)	1	1	0	0	2
Brassicaceae (1 species)	*Cardamine fallax* (4)	1	1	0	0	2
Phytolaccaceae (1 species)	*Phytolacca americana* (2)	0	2	0	0	2
Moraceae (1 species)	*Morus alba* (4)	0	1	0	1	2
Portulacaceae (1 species)	*Portulaca oleracea* (25)	0	0	0	2	2
Violaceae (1 species)	*Viola mandshurica* (3)	1	0	0	0	1
Commelinaceae (1 species)	*Commelina communis* (7)	0	0	0	1	1
Convolvulaceae (1 species)	*Calystegia pubescens* (1)	0	0	0	1	1
Equisetaceae (1 species)	*Equisetum arvense* (6)	0	0	0	1	1
Araceae (1 species)	*Pinellia ternata* (19)	0	0	0	1	1
27 families	56 species (366)	56 (38.4)	51 (34.9)	1 (0.7)	38 (26.0)	146 (100.0)
Unidentified (1 species)	Unidentified (5)	1	1	0	0	2
Total (371)		57 (38.5)	52 (35.1)	1 (0.7)	38 (25.7)	148 (100.0)

The values in parentheses indicate the number of detected infections, expressed as percentages, for each virus type. For mixed virus infections, each virus was counted individually.

**Table 7 viruses-17-00383-t007:** Analysis results of the correlation between *P. edulis* viruses and infecting weeds surrounding the cultivation sites.

Classification ^z^	P_CMV	P_PaLCuGdV	P_EuLCV	P_EAPV	W_CMV	W_PaLCuGdV	W_EuLCV
P_PALCuGdV	0.23 **						
P_EuLCV	0.23 **	0.57 **					
P_EAPV	0.07	0.12	0.33 **				
W_CMV	−0.00	0.13 *	0.12	−0.04			
W_PaLCuGdV	−0.00	−0.00	0.07	0.11	0.06		
W_EuLCV	−0.00	0.06	0.09	0.16 *	0.14 *	0.61 **	
W_EAPV	−0.00	0.44	0.06	−0.00	−0.00	0.13	0.15 *

^z^ P_ and W_ indicate virus-infected passion fruit and weeds, respectively. * *p* < 0.05 and ** *p* < 0.01. Values indicate Spearman’s rank correlation coefficients.

## Data Availability

The original contributions presented in the study are included in the article. Further inquiries can be directed to the author.

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
