# Peer review of "Integrating Viral Infection and Correlation Analysis in Passiflora edulis and Surrounding Weeds to Enhance Sustainable Agriculture in Republic of Korea"

_viruses, 2025, doi:10.3390/v17030383_

Round 1

Reviewer 1 Report

Comments and Suggestions for Authors

Review report for the paper entitled “Integrating viral infection and correlation analysis in Passiflora edulis and surrounding weeds to enhance sustainable agriculture in South Korea”

The present study presents the prevalence of viral infections in commercial cultivations of passion fruit in South Korea. The study is well planned and executed with samples taken over a period of five years which shows the robustness of the data and prevalence of infecting viruses. It is commendable to watch a single person carrying out a study spanning years of sample collection and data analysis. The key message of this article is mentioned in lines 399-403. This study merits a publication in Journal “Viruses” as this data will not only benefit the groups working with Plant viruses but would also serve as a good example to a broader audience encompassing plant pathologists.

I would like the author to respond to some minor concerns:

  1. Introduction section mentions about the losses in PE by viruses, but there is no reference cited for this, please add appropriate reference.
  2. I would like the author to comment on whether the samples were collected from same plants or different plants during the repeat visits. If indeed there were some repeat samples, then I would suggest the author to have a relook into the data and can do a reanalysis wherein the author can find if there was progressive increase in number of viruses infecting a particular sample.
  3. If possible I would like the author to add some data on seasonal variation in virus detection.

Reviewer 2 Report

Comments and Suggestions for Authors

The MS by Choi presents PCR-based diagnosis of different viruses infecting passion fruit in South Korea. Author mentioned in Line 57 that such report is already available from South Korea (Ref. 17).  So, rationale of this study is not clear. PCR-based diagnosis is a routine test for detection of plant viruses, so the MS does not have any novelty/originality. Authors have performed only PCR-based diagnosis to conclude the presence of different virus species, however, cross-reactivity of the primers can not be ruled out. Authors did not show whether the primers they have used show cross-reactivity. Without such confirmation, conclusion of mixed infection can not be verified. So the content of the MS has little significance. Hence, the data generated is not scientifically sound. Authors could have shown detection of these viruses with another diagnostic assay like ELISA or sequencing to support the result of PCR-based diagnosis. The English is very poor and that resulted into too complex and irrelevant meaning of the sentences. Due to the poor sentence construction, presentation of the result is not sound. The MS is not suitable for publication.

Comments on the Quality of English Language

Poor English. Long complex poorly constructed sentences.

Reviewer 3 Report

Comments and Suggestions for Authors

This is a single-author manuscript. Is the author claiming they undertook all the experimental work and writing alone? Please explain

Do not shorten Passiflora edulis to ‘PE’. Adhere to scientific conventions for naming taxa

L16 define heterogeneous relationships

L17 ‘occurring alongside other viruses’ is replaced with coinfected by other viruses

L21 define the ‘actionable strategies’ recommended by your study.

L28-29 Two subspecies…

L31, 32 Use metric units throughout

L47 A better reference is Wu W, Ma F, Zhang X, Tan Y, Han T, Ding J, Wu J, Xing W, Wu B, Huang D, Zhang S, Xu Y, Song S. Research Progress on Viruses of Passiflora edulis. Biology (Basel). 2024 Oct 19;13(10):839. doi: 10.3390/biology13100839. PMID: 39452147; PMCID: PMC11506102.

L66 Vertical and horizontal transmission

L78 ‘However, to our knowledge, no research group has investigated the relationship between the viral infection of PE and that of weeds growing surrounding the PE cultivation sites, which can serve as useful information for preventing the infection and spread of the viral diseases in PE cultivation sites. ‘

Author, Did you thoroughly investigate the literature? Example: Junco, M.C., Silva, C.D.C., do Carmo, C.M., Kotsubo, R.Y., de Novaes, T.G. and Molina, R.D.O., 2021. Identification of potential host plants of Cowpea aphid‐borne mosaic virus. Journal of Phytopathology, 169(1), pp.45-51.

L95 PE in Jeollabuk-do (Jeonbuk State)…

L97-99. The areas seem small. Check they are correct.

L105. Were the plants collected randomly or only symptomatic? Which plant species?

Table 1. Why were tests for only 5 viruses used? What were the annealing temperatures of the primers?

L154 ‘Weed surveys targeted all types of weeds surrounding PE cultivation sites from 2018 to 2021.” Which species of weeds were collected, how many of each and how were they chosen? 283 passion fruit samples were collected. How many weeds?

Results and discussion are usually two distinct sections.

Comparing your results with others would be easier to follow in a table

L213 And full genome analysis

Table 2 has a column ‘uninfected’ How were these proven to be uninfected? Or do you mean not infected with the 5 viruses tested for?

Tables 2 and 3. I don’t understand why decimals are used in the rates with such small samples. I recommend you round up or down to whole numbers.

Table 6 does not show the number of plants tested

Aphids are mentioned by no data given

Round 2

Reviewer 2 Report

Comments and Suggestions for Authors

Manuscript can be accepted now.

Reviewer 3 Report

Comments and Suggestions for Authors

The amendments and responses are appropriate. I wish the author all the best to continue this valuable work.